# A Review on Nanoparticles as Boon for Biogas Producers—Nano Fuels and Biosensing Monitoring

**Shah Faisal [1], Fauzia Yusuf Hafeez [2], Yusuf Zafar [3], Sabahat Majeed [1,2], Xiaoyun Leng [1], Shuai Zhao [1], Irfan Saif [1], Kamran Malik [1] and Xiangkai Li [1,*]**

[1]  Ministry of Education Key Laboratory of Cell Activities and Stress Adaptations, School of Life Sciences, Lanzhou University, Tianshuianlu≠ 222, Lanzhou 730000, China; fs15@lzu.edu.cn (S.F.); zeypher_saba@yahoo.com (S.M.); lengxy09@imust.cn (X.L.); zhaosh15@lzu.edu.cn (S.Z.); irfanmicro66@hotmail.com (I.S.); malik15@lzu.edu.cn (K.M.)

[2]  Department of biosciences, COMSATS University Park Road, Tarlai Kalan Islamabad, Islamabad 45550, Pakistan; fauzia@comsats.edu.pk

[3]  Pakistan Agriculture Research Council (PARC), Ataturk Avenue, G-5/1, Islamabad 44000, Pakistan; y_zafar@yahoo.com

*  Correspondence: xkli@lzu.edu.cn; Tel.: +0931-8912560

**Abstract:** Nanotechnology has an increasingly large impact on a broad scope of biotechnological, pharmacological and pure technological applications. Its current use in bioenergy production from biomass is very restricted. The present study is based on the utilization of nanoparticles as an additive to feed bacteria that break down natural substances. The novel notion of dosing ions using modified nanoparticles can be used to progress up biogas production in oxygen free digestion processes. While minute nanoparticles are unstable, they can be designed to provide ions in a controlled approach, so that the maximum enhancement of biogas production that has been reported can be obtained. Nanoparticles are dissolved in a programmed way in an anaerobic atmosphere and are supplied in a sustainable manner to microbiotic organisms responsible for the degradation of organic material, which is a role that fits them well. Therefore, biogas fabrication can be increased up to 200%, thereby increasing the degradation of organic waste.

**Keywords:** biofuels; bio-methane; environment; nanoparticles; nanotechnology; biosensors; waste activated sludge

## 1. Introduction

Over the past few decades, industrialization and population growth has led to a significant increase in energy demand. Currently, fossil fuels are the prime source of basic energy production, contributing 80% of total global consumption. Out of this 80% of primary energy produced by fossil fuels, the transport sector is the major consumer with 58% consumption [1,2], of which 80% is being produced by Brazil and USA [3]. In future, the transportation fuel demand is estimated to increase up to 55% globally by 2030 and this will increase the demand for biofuels [3–5].

Due to this intensive consumption and increasing demand in the energy sector, fossil fuel resources are depleting at a rapid pace and there is a dire need to explore and identify new and renewable energy sources globally [6].

One such renewable energy source is biogas produced by anaerobic digestion (AD), which utilizes various wastes such as animal manure, [7] agricultural waste [8] and organic wastes [9]. Biogas is produced mainly due to the process of AD, resulting in the formation of $CO_2$ as a byproduct, which is consumed during photosynthesis and retrieved again, for AD, in the form of agricultural waste and animal manures. This consumption of $CO_2$ takes place in a closed cycle [10], as shown in Figure 1.

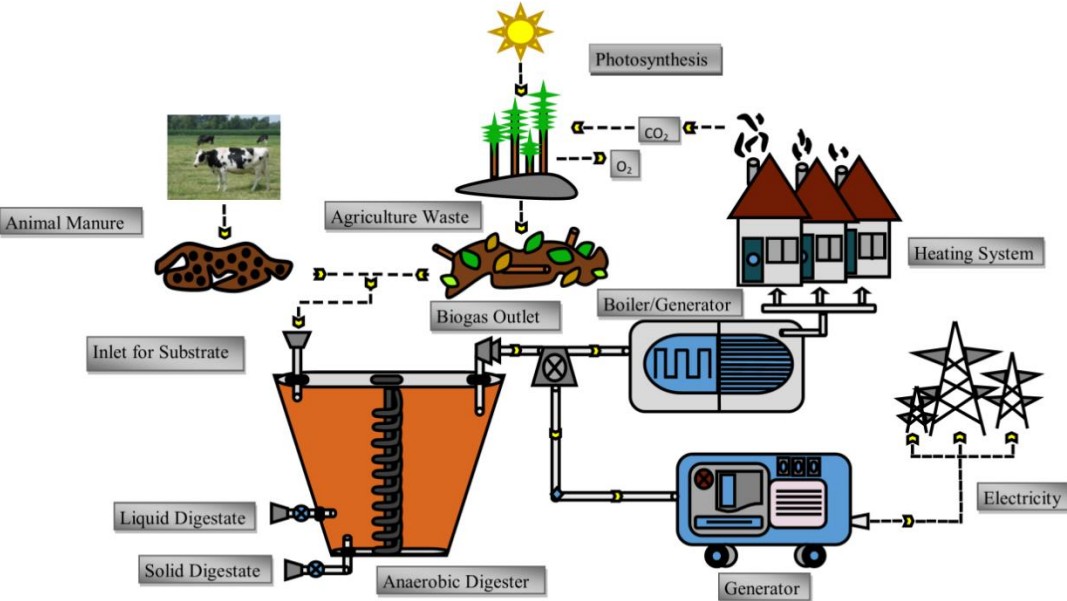

**Figure 1.** Waste utilization to produce renewable energy.

During AD, four steps are involved in methane production, which include; hydrolysis, acidogenesis, acetogenesis, and methanogenesis. Methane production is a result of the synthrophic microbial relationship. During hydrolyses, bacterial cellulosome and exoenzymes monomerize complex proteins, carbohydrates and fats. In the second step (acidogenesis), along with $CO_2$, hydrogen and alcohols, further degradation of monomers into short chain acids takes place. In the third step (acetogenesis), short chain acids are converted into acetate, $CO_2$ and hydrogen. In the last step (methanogenesis), intermediates are converted into $CO_2$ and methane by methanogens [11] (Figure 2).

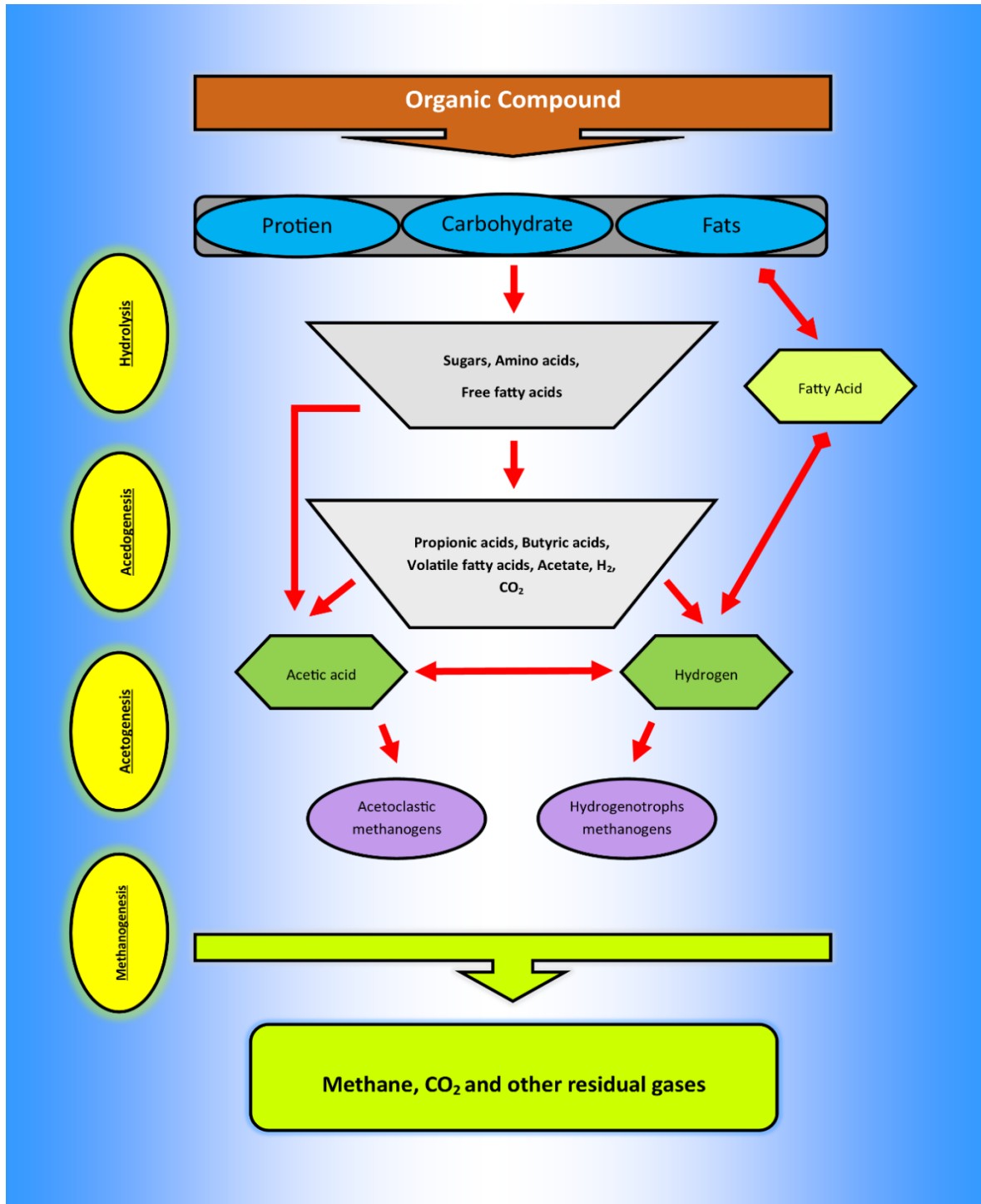

**Figure 2.** $CO_2$ and biomethane formation in an anaerobic process.

## 2. Application of Biosensors in Biogas Monitoring

The assessment of anaerobic digestion is based on the continuous monitoring of organic and volatile fatty acids, resulting in the accumulation of intermediates for unsteady progression conditions [12]. The intensifying public interest in biogas production is a result of the exhaustion of fossil fuels. Anaerobic digestion has the advantage of exploiting industrial waste for energy production and thus treating another modern day problem [13,14]. Efficient methane production and endurance of process stability are resulting outcomes based upon the improvement of several

economic and technological aspects. These include a suitable feedstock composition, appropriate biogas purification technologies and ideal conditions for a biogas reactor, which is based upon several physical and biochemical parameters, including pH, alkalinity, gas quality, FOS/TAC (Flüchtige Organische Säuren, i.e., volatile organic acids/Totales Anorganisches Carbonat, i.e., total inorganic carbonate) [15–17]. The accumulation of organic acids like formate, lactate and alcohols, and volatile fatty acids (e.g., propionate, acetate, butyrate) results in acidification of the reactor, which clearly indicates process imbalance [18–21]. The conventional methods for estimation of acid composition are gas chromatography [22], spectroscopy [23,24] and HPLC (high-performance liquid chromatography) [25,26], which are commonly carried through external sources that cause high cost partanalysis.

*Determination of Different Products in AD*

Instantaneous intervention is not possible due to the time between sampling and the availability of the results, and hence, uncertainty among the plant operators occurs. Acid content is typically only analyzed once or twice per month. To combat this problem, biosensors have been devised for accurate and fast analysis of many compounds. Ethanol and lactate biosensors have had more attention due to their vast applications in the food and healthcare industry [27,28]. The correlation between these intermediates and process stability has resulted in many studies being carried out for the optimization of and development of an organic acid (OA) biosensor, comprising enzyme production for specific detection of formate, ehanol and D/L lactate, in contrast to the limited concentration of the volatile fatty acid (VFA) biosensors [29–32]. These analytes are detected through microbial electrolysis cells [33], microbial fuel cells [32] or dissolved oygen probes with an immobilized biofilm [34], while enzyme-based sensors have been used for the peculiar determination of individual substrates, like acetate and propionate [35–37].

Furthermore, issues like poor biodegradability, unstable fermentation and low methane production still exist with proper utilization of this technology [38]. To overcome these issues and for efficient energy production, various technologies have been used in different studies, such as pre-treatment of sludge [39], addition of various substrates to the sludge [40] and introduction of different additives to the digestive system.

Nanotechnology is an incipient technology for scientific development and holds significant prospects for improving environmental protection technologies. Water pollution, air pollution, and disproportionate consumption of natural resources are some of the formidable challenges faced by communities globally. The application of nanotechnology to environmental protection is receiving more attention and is being applied in the areas of water treatment, biomass pretreatment, biohydrogen production, wastewater treatment, groundwater remediation, soil remediation, waste management, etc. (Figure 3). The volume of nanomaterials and nano-products is anticipated to grow dramatically soon and the effective management of nano-waste is of great concern. Due to the specificity in their physical and chemical properties, nanoparticles are mostly used in industrial and consumer products. They are adaptable remediation materials because they are moderately reactive in water and can also serve as an excellent electron donor [41].

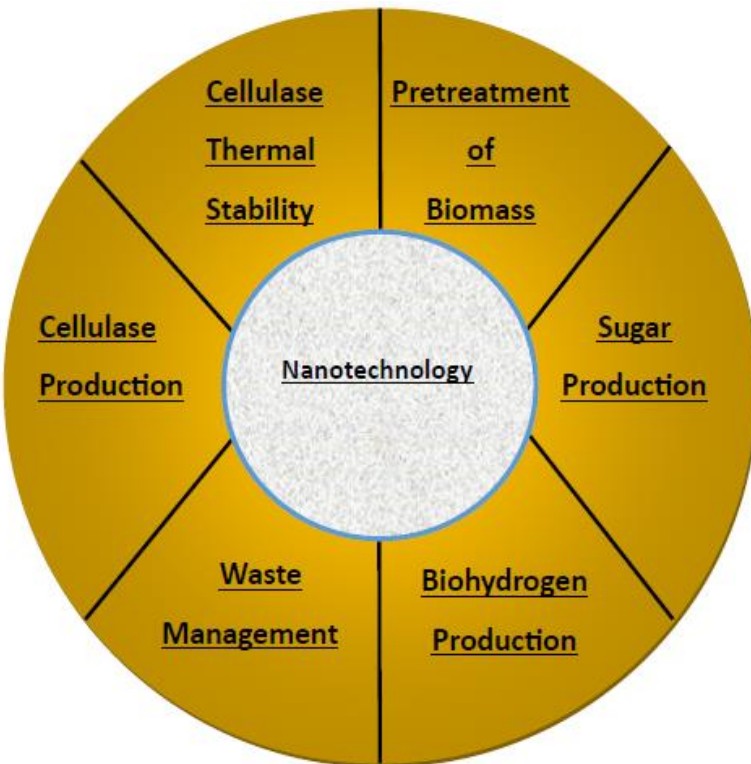

**Figure 3.** Various applications of nanotechnology.

Removal of waste contamination using nanoparticles has been studied for the last two decades. Biodegradable nutrients present in municipal sludge used to be regarded as a biomass resource. AD is used to dispose of the biomass, which results in mitigation of pollution along with the generation of renewable biogas energy. Efficiency and stability of AD has been enhanced significantly over the past years, resulting in a significant reduction in disposal and transportation costs. [42–44].

The activity of organic matter-degrading bacteria is enhanced when fed with nanoparticles, resulting in high production of biogas. Due to the high rate of biogas production, most of the waste is degraded and utilized, while the remaining waste can be utilized as an improved compost due to iron enrichment. It is novel to use anaerobic digestion processes for improving biogas production by dosing iron ions consuming engineered nanoparticles. Due to the fluid nature of minute nanoparticles, the output of ions can be projected in a controlled way, which results in the highest biogas production ever reported [45].

Efficient energy production in all industries and economically leveraged renewable energy outputs can be enhanced through nanotechnologies. They can enhance the transformation of biomass to chemical intermediates, specialty chemicals, fuels and products, and can be utilized as an important instrument for increasing the efficiency of bioenergy. Nanomaterials can interact with some species of algal biomass and active sludge. For satisfying the energy demands of the world, algal biomass has been widely predicted to be the future energy source. Macroalgae, such as seaweed and water hyacinth, interact substantially with nano-molecules. The concentration of $CH_4$ and biogas production is enhanced by directing modified nano-particles to the optimization of anaerobic digestion processes. Thus, the optimization of the process of energy production to a larger or commercial scale is limited only by the gain of a low dosage of nano-molecules [46].

The synthesis of nano-structured nanoparticles is carried out mostly by wet chemical methods, which include surfactant mediation, electro-deposition, emulsion, precipitation, etc., in this study. The rate of methane production is raised by using nanoparticles in an anaerobic digester. Due to their toxicity, it may not be possible to add particles directly. Nanoparticles are added to overcome this problem. Conventional biogas production can convert only 30 to 40% of the organic matter, which is

not very effective in comparison with energy sources. Thus, for the improvement of biogas production, there is a desperate need for modern approaches. Iron strongly enhances anaerobic digestion, but its function in a standard configuration is debatable in an anaerobic closed reactor. However, improved nano-molecules are able to slowly dissolve biodegradable nano-particles and avoid various problems such as bacteriostaticity by facilitating the production of ions in the reaction medium. [47]

This work is grounded on the evaluation of physical parameters for finding a path to increase the yield rate of biogas. Better outcomes are shown by nanoparticles in comparison with their bulk form. There are few metal oxide particles in the literature that enhance the quality and yield of biogas [48]. On the nano scale, great potential is exhibited by nanoparticles as catalysts, absorbents for wastewater treatment, coatings, ion exchangers, gas sensors, flocculants, pigments, magnetic recording devices, toners, magnetic information storage devices, xerography inks, magnetic resonance imaging (MRI), medicine and bio-separation. The latest studies of nanoparticles suggest an increase of 40 to 200% in biogas production by using nano-molecules in the digestive [49]. The 100 ppm of 7 nm nanoparticle was used, resulting in an increase of 180% in biogas production. The deductive reasoning and applications of nano-structured oxides/hydroxides have been reported by Mohapatra and Anand (2010) [50].

## 3. Metallic Nanoparticles Used for the Enhancement of Bio Gas Production

### 3.1. Nanoparticles

'Nanomaterials' are the materials with an external dimension or internal or surface structure on a nano scale ranging from 1 to 100 nm in size [47,49]. The chemical origin of nanoparticles is greatly influenced by their chemical origin, which is responsible for their behavior and fate in the environment [51,52]. Nanoparticles are classified into four groups: organic, inorganic, composite and carbon NPs. Nanoparticles possess special chemical, physical and optical characteristics. At the nano scale, properties of the particles change unpredictably, making them behave differently with the same substance at the macro scale. Nanoparticles are ideal in a diversity of areas, such as energy, medical, electronic and commercial products, due to their high reactivity and special features. Using nanoparticles leads to the production of efficient, durable, lighter, firmer, and cleaner products and materials [53].

Different chemical and physical properties of nanoparticles from their macro counterparts make them interesting. The higher chemical reactivity of nanoparticles is due to their high surface area, providing a greater number of reaction sites [54]. Gold (Au) is another example of nanoparticles at the nano scale. Amber does not react with many chemicals at the macro scale and behaves as an inert element, but at the nano scale, gold becomes enormously reactive, behaving as a catalyst to speed up reactions [54]. This extremely reactive property of nanoparticles is due to the ratio between the mass and open area. The human digestive system is a biological example of AD processes being determined by the surface area to volume ratio, microorganism activity aids AD digestion.

### 3.2. Concentration of Nanoparticles

Nanoparticles have been acquired from both anthropogenic and natural resources. In waste sludge, a very high concentration of NPs could have accumulated. However, the toxicity and the impact of NPs on the sludge treatment stream is still an area that requires a great deal of research [55]. Nguyen determined the effects of ZnO NPs and $CeO_2$ nanoparticles on the sludge AD process, toxic potential of sludge to plants and bacteria and dewatering process of the sludge.

The concentration of nanoparticles is very important in determining their role for the process of methane and biogas production (Table 1). Not all nanoparticles stimulate the anaerobic digestion system, rather some nanoparticles inhibit the production rate considerably when compared with a controlled sample. Types and concentration of nanoparticles play a vital role in the production rate of the anaerobic digestion system. In comparison with a control sample, the exposure concentration

of ZnO at 1000 mg/L resulted in inhibition to 65.3% biogas volume and 47.7% methane composition. At an endurable exposure concentration of zinc oxide, the inhibition effect could be overcome after an incubation of 14 days [56].

**Table 1.** Nano additives concentration and their impact on the biogas and methane production rate.

| NPs Type | NP size | Concentration | Feedstock | Temperature | Incubation Time | Effect |
|---|---|---|---|---|---|---|
| $CeO_2$ | 192 nm | 10 mg/L | Sludge from UASB reactor | 30 °C | 40 | 11% Increase in biogas production [55] |
| $Fe_3O_4$ | 7 nm | 100 ppm | Waste water Sludge | 37 °C | 60 | 180% Increase in biogas production and 234% increase in methane [46] |
| $Fe/SiO_2$ | | 105 mol/L | – | 55 °C | – | 7% Increase in methane production [57] |
| $Pt/SiO_2$ | – | 105 mol/L | – | 55 °C | – | 7% Increase in methane production [57] |
| $Co/SiO_2$ | – | 105 mol/L | – | 55 °C | – | 48% Increase in methane production [57] |
| $Ni/SiO_2$ | – | 105 mol/L | – | 55 °C | – | 70% Increase in methane production [57] |
| Co | 28 nm | 1 mg/L | fresh raw manure | 37 °C | 40 | 71% increase in biogas production / 45.92% increase in methane production [58] |
| Ni | 17 nm | 2 mg/L | fresh raw manure | 37 °C | 40 | 78.53% increase in biogas production / 116.76% increase in methane production [58] |
| Fe | 9 nm | 20 mg/L | fresh raw manure | 37 °C | 40 | 47.7% increase in biogas production / 67% increase in methane production [58] |
| $Fe_3O_4$ | 7 nm | 20 mg/L | fresh raw manure | 37 °C | 40 | 73% increase in biogas production / 115.66% increase in methane production [58] |
| ZnO | 140 nm | 1 mg/g-TSS / 10 mg/g-TSS / 50 mg/g-TSS | WAS AGS | 35 °C | 40 / 105 | No effect [59] / No effect [60] / No effect [60] |
| nZVI | <50 nm | 10 mg/g-TSS | WAS | 37 °C | 30 | 120% increase in methane production [61] |
| $Fe_2O_3$ | <30 nm | 100 mg/g-TSS | WAS | 37 °C | 30 | 117% increase in methane production [61] |

Luna del Risco reported the AD process of cattle manure and effects of metal oxide on methane and biogas productivity [56]. During the experiment, the influence of CuO nanoparticles was higher than the other test compounds. At day 14, out of the total biogas produced in the control, there was a reduction of 30% caused by 15 mg/L concentration of Cu. CuO in bulk, with concentrations of 120 and 240 mg/L, caused a reduction by 19% and 60%, respectively, whereas in the presence of Cu microparticles, biogas production was less inhibited. Differences between the bulk particles and nanoparticles of CuO ($p < 0.05$) were validated by statistical analysis. Heinlaan [62], Neal [63] and Kasemets et al. [64], reported that a toxicity of nanoparticles to bacteria, causing damage to the cell membrane and inhibiting biogas production by the release of metal ions. Whereas, ZnO bulk particles were compared with test samples containing ZnO nanoparticles. An inhibition of 74% and 43% of the highest yield resulted from ZnO nanoparticles concentrations of 120 and 240 mg/L, respectively. Test bottles containing bulk ZnO showed a decline of 18% and 72% of the total biogas yield at day 14. Both bulk and nanoparticles of ZnO were unable to produce a prominent difference in biogas inhibition [56]. From the above mentioned observations, it can be concluded that biogas yield is affected by CuO and ZnO nano-sized concentration and particle size.

$CeO_2$ can enhance biogas production by 11%, even at the low concentration of 10 mg. A bacterial toxicity test also confirmed a positive effect of $CeO_2$ at a similar concentration. However, the toxicity levels of both the nanoparticles decreased when they were applied to the sludge, rather than naturally occurring. Bacterial toxicity also reduced when the AD process ended. In addition, the exposure concentration of NPs was directly proportional to the time required for dewatering the digested sludge. After the end of the AD processes, toxicity of the sludge also reduced notably. After exposure of the sludge to a concentration of 1000 mg/L of $CeO_2$ nanoparticles, there was an inhibition of 47.5% for bacterial viability, but after the AD process, the sludge had an inhibition of 30.4% for bacterial viability. Similarly, 1000 mg/L of ZnO nanoparticles before digestion had an inhibition of 92.3%, while this value was just 34.8% after the AD process [55].

Due to the presence of $Fe_3C$ and non-toxic $Fe_3C$ ions, methane production can be enhanced by a nano iron oxide ($Fe_3O_4$ NPs). Nanoparticles of $Fe_3O_4$ at 7 nm were added to an anaerobic waste digester with a concentration of 100 ppm at 37 °C for 60 days. There was an increase of 234% in methane production and 180% in biogas production, which could be considered the highest and most remarkable increase in biogas production using nanoparticles [46]. Due to an enhancement in AD, higher organic matter processing and methane production can be achieved by utilizing a $Fe_3O_4$ (magnetite) nanoparticles delivery system. This improvement in performance is due to the presence of $FeC_2/FeC_3$ ions in the reactor. These ions are introduced in the form of nanoparticles, just like controlled drug delivery systems. As Fe plays a vital role in electron transport, methane and hydrogen production rates increase and bacterial growth is stimulated by promoting enzymatic actions [65]. Due to better biocompatibility and low toxicity, $Fe_3O_4$ nanoparticles are the most prevalent materials [66].

Abdelsalam et al. [58] carried out a study on fresh raw manure using Ni, $Fe_3O_4$, Co, Fe nanoparticles and found that after 40 days, Ni NPs with a concentration of 2 mg/L and a particle size of 17 nm were the most effective NPs, with highest average methane and biogas yields of 78.53% and 116.76%, respectively. While $Fe_3O_4$, with a particle size of 7 mm and a concentration of 20 mg/L, yielded a 73% increase in biogas production and 115.66% increase in methane production. Co, with a particle size of 28 nm and a concentration of 1 mg/L, exhibited a 71% increase in biogas production and a 45.92% increase in methane production. Fe NPs, with a particle size of 9 nm and a concentration of 20 mg/L, resulted in a 47.7% increase in biogas production.

In another study, Wang et al. [61] utilized Waste Activated Sludge (WAS) for methane production and applied nZVI (10 mg/g) and $Fe_2O_3$ (100 mg/g) NPs, and found an enhancement of 120% and 117%, respectively. These results reveal that lower concentrations of said NPs stimulate the microbial populations of Archaea and bacteria.

## 4. An Understanding of Biomass and Their Characteristics

Biomass resources were analyzed on the basis of chemical, biological, physical composition and in terms of their source. Evaluation of the impact of biomass particle size on the biomass to bioenergy conversion basis was carried out, in both biothermal and biochemical aspects. Different biomass types were studied based on pretreatment and particle size. In terms of structure and composition, different effects were produced by different pretreatment methods [67]. For example, lignin can be removed from alkaline pretreatments, and the biomass hemicellulose fraction can be removed by acidic and biothermal pretreatment. Reduction in particle size to increase the biomass specific surface can be carried out to reduce cellulose fiber organization during biomass fibrillation by a milling-based pretreatment, measured by a decline in crystallinity.

Enhancement of economically leveraged renewable energy production and energy efficiency can be done with the help of nanotechnology. Nanomaterial interactions were evident with a few algal biomass species and active sludge. With regards to bioenergy production inhibitory [55], an adverse or increased yield [68] was visible. Particle surface area to volume ratio, leading to variation in the severity of the effects, was also assessed. NPs' effects on energized sludge systems. The outcomes of the NPs impact on energized sludge can be seen in a few related characteristics of NPs and energized sludge. These include aggregation, size of nanoparticles and reciprocation of microorganisms in its bioenergy production.

*Interaction of Nanoparticles with Biomass*

Many nanoparticles have antimicrobial properties, and can therefore play a vital role, with liquid biomass, in water purification [69]. Enhancement of the AD of iron is summarized in (Table 2. Nowadays, NPs are being used for the detection and separation of biological and chemical substances, such as metals (Zn, Cd, Cu, etc.), algae (cyanobacterial toxins, etc.), nutrients (ammonia, nitrate, phosphate, etc.), cyanide, antibiotics, organics, parasites, bacteria and viruses. For water purification, four classes of nanomaterials are being evaluated as functional materials: dendrimers, zeolites, NPs

containing metal, and carbonaceous nanomaterials. Nanofibers and carbon nanotubes (CNTs) also show an affirmative result. Due to a high surface area to volume ratio, better results have been revealed by nanomaterials than other water purification techniques [70]. Monitoring, adsorption, microbial control, photocatalysis, membrane processes, disinfection, monitoring and sensing are potential and current applications of nanotechnology for wastewater and water treatment [71]. But facts and data related to the toxicity of nanomaterials are still not sufficient [72]. There are two main factors responsible for the antibacterial activity of nanoparticles: (i) the physiochemical properties of nanoparticles, and (ii) the nature of bacteria. It has been observed that the antibacterial effect of silver nanoparticles are enhanced by treating coliform bacteria with ultrasonic radiations for a short period prior to nanoparticle treatment. Ag NPs also displayed major activity against biofilms of bacteria. The antibacterial effect, and Ag NPs concentrations, depend upon the class of bacteria to be treated [73].

Previous research has demonstrated that *Vibrio cholera* and *Pseudomonas aeruginosa* showed more resistance than *Salmonella typhi* and *Escherichia coli*, but bacterial growth was completely abolished at concentrations above 75 mg/L [74,75]. The antimicrobial activity of silver NPs against *Staph aureas* and *Escherichia Coli* has also been studied, and at low concentrations, *Escherichia coli* was inhibited but *Staph aureas* was less inhibited [76]. Ag NPs also showed significant adverse effects on filamentous green algae [77]. The biomass to bioenergy conversion by the interaction of nanoparticles has shown the importance of nanomaterials in research. This conversion could be either a biological, chemical or thermal process. The conversion process is affected by inorganic contaminants obtained from organic biomass and molecular size. Functionalized nanoparticles are acquired from both natural and synthetic sources. Within the process, a number of complications can arise because of the existence of multi-faceted interactions, therefore, it needs to be tackled properly. Two types of biomass, i.e., waste sludge and algae, describe the impact of NPs on biomass energy conversion.

**Table 2.** Summary of anaerobic digestion enhanced by nanoparticles.

| Substrate | Iron Type | Dosage (g/L) | Temperature (°C) | Increment in CH$_4$ Yield (%) | Increment in COD/VSS Removal (%) | Reference |
|---|---|---|---|---|---|---|
| Excess sludge | Scrap iron | 10 | 35 | 10.1 [a] <br> 21.4 [b] | 83.3 | [78] |
| Excess sludge | Scrap iron | 4 | 35 | 43.5 | 33.6 [c] | [79] |
| Swine wastewater | ZVI powder | 25 | 30 | 145.5 | 56.2 [d] | [80] |
| Excess sludge | nZVI | 1 | 37 | 25.2 | 22.0 [d] | [81] |
|  | ZVI powder | 16.7 | 37 | 40.8 | 48.4 [d] |  |
| Excess sludge | mZVI | 10 | 35 | 131.6 | NA | [82] |
|  | nZVI | 10 | 35 | 46.1 | NA |  |
| Pig manure | mZVI | 20 | 35 | 20 | NA | [83] |
| Manure | nZVI | 0.02 | 37 | 159 | NA | [84,85] |
|  | mZVI | 12 | 35 | 41.7 | 105.9 [c] |  |
| Excess sludge | Scrap iron | 2.385 | 35 | 38.3 | NA | [86] |
| Excess sludge | nZVI | 16.6 | 35 | 1304 | NA | [87] |
|  | Scrap iron | 33.3 | 35 | 25.3 | NA |  |

ZVI-Zero-valent iron; nZVI-nano scale zero-valent iron; mZVI-microscale zero-valent iron; NA-not available; OFMSW-Organic fraction of municipal solid waste; [a] acidogenesis phase; [b] methanogenesis phase; [c] VSS; [d] COD.

## 5. Nanoparticles with Microorganisms

Nanoparticles have significant effects on microorganisms. NPs strain latent detrimental effects on wastewater microorganisms, according to an overview of their antimicrobial properties. Although, at present, statistical data on the NPs' effect on wastewater microorganisms during aerobic digestion are rather minimal, but it still has a remarkable effect [88,89]. Hence, it is tough to make a particular claim regarding the harmful effect of NPs on wastewater microorganisms. However, minimized efficiency of

AS and AD processes, absolute collapse of treatment and environmental pollution from contaminated effluents and utilization of biosolids for changes in soil texture may result due to NPs and microbial community contact [90].

A broad range of microorganisms are affected by the silver ion. Bacterial growth in a variety of medical treatments, including dental work, catheters, and healing burn wounds, has been controlled in recent days by silver ions [91]. Concentration and contact time influencing the mechanism of release of ions from Ag was exhibited by *Escherichia coli*. The leaking of reducing sugar and protein, enzyme inhibition, cell obstruction, and disperse vesicles, which slowly disintegrate, are the detrimental effects inhibiting cellular respiration and cell growth [92].

### 5.1. Algal Biomass and Nanoparticles

In order to fulfill the future energy requirements of the world, algal biomass is potentially the next energy resource for bioenergy production. It also has a potential to serve as a resource of high value for basic chemicals and extracts. Being a member of aquatic systems, they serve as food sources and hold a key role in photosynthesis. In fresh water, the impact of nanoparticles on microalgae can be found in the form of silver ions, silver chloride (AgCl) and silver sulfide ($Ag_2S$). Silver ion is the most toxic form of silver NPs [93]. Due to an increase in concentration in an aquatic environment, these nanoparticles can damage and affect the biota [88,94]. In drinking water, ground water and surface water, the concentration of silver nanoparticles has already been found to be above 5 g/L [95]. Among many of the possible reasons for toxicity of nanoparticles, three are very obvious: (i) a high dissolution rate due to a high surface area to volume ratio [94], (ii) aggregation behavior due to bioavailability, and (iii) coating with the stabilized organic material [96]. Becaro [97] carried out an investigation on the toxicity of Ag NPs stabilized with PVA (polyvinyl alcohol) for aquatic microalgae, such as *Daphne similis*, *Pseudokirchneriella* subcapitata algae and *Artemia Salina*. With a size range of 218 nm, Ag NPs in solution were well dispersed under dynamic light scattering measurements. Lower toxicity was shown by Ag NP against *P. Subcapitata* and *A. Salina* while high toxicity exhibited to *D. Similis*. A major component of global aquatic ecosystems is formed by *Chara vulgaris* and *Pithophora oedogonia*, which are predominantly members of photosynthetic eukaryotic algae. According to Das et al., prominent adverse effects of nanosilver on the morphology and growth of filamentous algae were illustrated on a concentration basis [98]. When *Algal Thalli* was exposed to higher concentrations of Ag NPs there was continued algal chlorophyll content depletion and disturbance in mitoses chromosomal instability. Dramatic alterations in the algal cell wall, characterized by pithophoral degradation and ruptured cell walls, was revealed in SEM micrographs when treated with nanoparticles.

### 5.2. NPs Impact on Macro algae/Aquatic Plants

Zada et al. [99] demonstrated from water hyacinth that the fermentative production of hydrogen and ethanol is a feasible and sustainable process. Production of ethanol and hydrogen was significantly affected by iron nanoparticles. Fermentative hydrogen production and ethanol production were enhanced by iron nano-particles. In addition to concentrations already present in dry plant biomass, the optimum iron nanoparticles concentration in fermentative hydrogen production and ethanol production was 250 mg/L and 150 mg/L, respectively. Of the plant biomass, maximum hydrogen yield was 57 mL/g, which comprises 85.50% of the theoretical maximum hydrogen yield. The maximum ethanol yield was 0.0232 mL/g of the plant biomass, which is 90.98% of the maximum theoretical yield. This study concluded that different types of nanoparticles accumulate in water hyacinth.

## 6. Modified NPs, Particle Size and Their Effect

Excessive use of nanotechnology resulted in large-scale manufacturing and utilization of engineered nanoparticles (ENPs) on an industrial level. In comparison with CNTs and Ag nanoparticles, natural nanoparticles exist in a massive amount in the environment. Since natural NPs are generated in an uncontrollable way, most of the research has related to characterization and focused on ENPs.

A mixture of ENPs containing ZnO (20 nm), AgO (20 nm) and TiO$_2$ (30–40 nm) were compared with their bulk metal salts to evaluate their effects against non-spiked activated sludge (control) [100]. This study was conducted using three pilot treatment plants on a pilot scale. In comparison with the control plant, the specific oxygen uptake rate (SOUR), specific to microbes, increased 200% by introducing both nanoparticles and bulk metal mixtures. Selective damage was shown by scanning electron microscopy (SEM) on some microbial cells. Furthermore, due to the presence of NPs, flock size of activated sludge also reduced, but sludge volume index (SVI) remained the same. Various environmental factors such as a sludge volume index (SVI), natural organic matter, pH, light, etc., can affect the behavior and fate of NPs in the environment [101]. Bioavailable, chemical and physical properties of releasing NPs can be affected by various influences in nature.

It is very important to assess the potential risk of nanomaterials and nanoparticles by scrutinizing the expected transformation and mobility, and their interaction with other materials [51]. The behavior and reactivity of the impact of NPs on terrestrial and aquatic media is determined by the shape and particle size [102]. For example, nanoparticles with a size less than 30 nm produced a more cytotoxic effect to *S. aureus* and *Escherichia coli* [103] than an 8090 nm particle size [104]. This study proves that an AgO particle size greater 30 nm is unable to inhibit the microbial process. Size less than 5 nm in suspension is of particular interest due to its capability of nitrification inhibition in AS [105]. As shown for AgO, along with particle size, shape also plays a vital role and they can exist in rod, triangle or spherical shape. Out of these three distinct particle sizes, the truncated triangle form of silver oxide produced the strongest antibacterial effect on *Escherichia coli* in both broth and agar cultures [106]. There is no evident conclusion of this observation from pure culture to complex wastewater because microbial cells' interaction with NPs can be attenuated or enhanced by wastewater components.

## 7. Phytotoxicity/Ecotoxicity Effect of NPs

The effect of nanoparticles on inhibition in relation to the performance of AD needs to be investigated. Moreover, the sludge after digestion through AD is applied as compost and soil conditioner by dewatering. However, this sludge becomes inappropriate and toxic to apply as a biosolid due to the accumulation of nanoparticles. Therefore, digested sludge containing nanoparticles should be assessed in relation to phytotoxicity, and bacterial toxicity should be assessed before reusability of waste sludge. Moreover, further studies are required to find out the effect of nanoparticles on the dewatering ability of digested sludge. Therefore, it is not obvious whether nanoparticles can hinder the dewatering ability of digested sludge, eliminating of nanoparticle toxicity during AD, or can cause inhibition effects on plants and bacteria. These questions still needed to be answered [107].

There has been a significant focus on aquatic, rather than terrestrial, plants in term of ecotoxicity. Toxic effects of nanoparticles, on the germination and root growth of some plant species, have been reported in some studies [108]. One of these studies was designed to compare the effects of five types of commonly used nanoparticles with their corresponding bulk material in regards to biomass, germination and root elongation in the Cucurbita pepo (zucchini) plant. These particles included multi-wall carbon nanoparticles (MWCNTs), ZnO, Si, Ag and Cu. To ensure observation of relevant phytotoxic responses, 1000 mg/L was selected. A dose response study determined the effect of nanoparticles or bulk Ag concentration on transpiration, Ag content and biomass of the zucchini plant. Assessment related to the impacts of nanoparticles on agricultural plants will help find out the potential hazards of food chain contamination for human exposure as well as ecological exposure risk related to nanoparticles [109].

## 8. Mechanism of Microbial Activity

Nanoparticles are toxic to human and other living organisms. Due to their nano scale, they are easily exposed to humans and other organisms through ingestion, inhalation and dermal contacts. A large number of publications by various authors is available on the behavior, characterization and toxicological information of nanomaterials [110]. The focus of this research is mainly on

manufacturing of commercialized nanomaterials, which are widely applicable, such as fullerene, metal oxides and CNTs. It is important to assess the fate of nanopartical on the environment when applying them commercially on a larger scale. Ag nanoparticles are capable of interacting with the surface of various bacterial cells. This is especially relevant when dealing with Gram-negative bacteria because accumulation and adhesion of Ag NPs to the surface of bacteria has been observed in numerous studies. By damaging cell membranes, Ag NPs lead to structural changes, making bacteria more permeable [111]. This change is directly related to concentration, shape and size of the nanoparticles [112]. This influence is confirmed by a study using *Escherichia coli* that affirmed that gaps in the integrity of the player are created by the accumulation of Ag NPs, which increased the permeability leading to cell death of bacteria [113].

Due to their widespread medical, military and industrial applications, metal oxide nanoparticles, such as zinc oxide (ZnO), silicon dioxide ($SiO_2$), aluminum oxide ($Al_2O_3$) and titanium dioxide ($TiO_2$), have gained a lot of interest. They affect soil, aquatic organisms and human health upon their release into the environment. So far, the mechanism of toxicity for each nanoparticle is not understood exactly, but various characteristics may result in damage to the exposed organisms. Reactive oxygen species (ROS), such as super oxides ($O_2^-$), singlet oxygen ($_1O_2$) and free radicals ($OH^-$), are generated by nanoparticles that employ various adverse effects on microbes, such as scattered vesicles, disruption of cell wall, enzyme inhibition and protein and sugar membrane leakage leading to slow dissolution and resulting in inhibition of cellular growth and respiration [92,102]. During some studies, the metal oxide nanoparticles, ZnO, $SiO_2$ and $Al_2O_{3,}$ were proven harmful to *Pseudomonas fluorescens*, *Bacillus subtilis* and *Escherichia coli* [114]. Significant toxicity was caused by these nanoparticles to the viability of Gram negative bacterial cells by increasing their antibacterial effects. Chen et al. [115] reviewed the toxicity of nanomaterials on biomass and found that the chemical stability of nanoparticles of Ag, $TiO_2$, $Al_2O_3$ and $SiO_2$ have no adverse effects on microbes under anaerobic conditions, while Au nanoparticles showed low or no toxicity on microbes in anaerobic digestion, and $CeO_2$ nanoparticles presented the highest toxicity to both thermophilic and mesophilic microbes. As a result of metal ion release due to dissolution and corrosion of nanoparticles, the AD process was taxed. These toxic compounds can lead to obstruction of methane formation, a decrease in the methane content of biogas, or a complete failure of the process of methanogenesis.

## 9. Efficacy and Impact of Nanomaterials on Biomass

According to Theivasanthi et al., [116] nanoparticles synthesized through electrolysis exhibit antibacterial activities against both *Gram Positive bacteria* and *Gram Negative bacteria*. Antibacterial activity of copper is enhanced by an alteration in surface area to volume ratio. Antibacterial activity of copper NPs synthesized by electrolysis is higher than copper NPs synthesized by using a chemical reduction method against *Escherichia coli*. When using electric power for the synthesis of copper NPs, antimicrobial activity is increased. Availability of the chemicals required for the synthesis of NPs is easily accessible, cheap and non-toxic. Minimum infrastructure is required for the technology implementation. It is experimentally proven that this material could be used in antibacterial packaging, water purification, air filtration, air quality management, etc. For biochemical conversion of biomass, microorganisms play a vital role.

Various nanomaterials affect the performance of AD. The mechanism of interaction of these particles is very complex and the way they interact with the biomass, the process of conversion, overcoming the adverse effects and optimizing the positive effect, all need to be understood better. The rate of AD can be influenced by the particle size because it affects the surface area for biomass biodegradation. Regardless of the chemical constituents, all nanoparticles possess an extremely high surface area to volume ratio. Therefore, the atoms of the surface and capping agents dominate the physical properties of nanoparticles. For applications such as catalysis, a high surface area to volume ratio is vital. The greater the surface of the same material, the greater is the reactivity.

Different nanoparticles stimulate different responses and interactions among microorganisms. Although a few studies concluded that copper NPs have significant potential as bactericidal agents, NPs such as iron oxide, silica and its oxides, gold and platinum have not shown a bactericidal effect in *Escherichia coli* studies. By stimulating the bacterial growth, magnetite NPs ($Fe_3O_4$ NPs) can enhance methane production. Another application of NPs is as a fuel catalyst for the reduction of harmful engine combustion emissions. Recent studies found that the AD process, biodegradation and nitrification are inhibited by NPs [107,117]. The particle size, time and concentration determine whether there is enhancement, inhibition or adverse effects of energy conversion.

Biomass, such as municipal solid waste and agriculture, still possesses a huge potential scope in relation to nanomaterials. It is very important to find out the best possible use of nanoparticles in bioenergy systems. This review could serve as an important tool for the future in order to carry out further research on 'nanotechnology in bioenergy'.

## 10. Recommendations and Conclusions

Efforts are being made globally to cut down the emission of carbon in order to reduce its impacts on global warming and the emphasis is on the carbon-neutral operation of biofuels. Production of methane is obtained by converting organic matter contained in sludge through the process of wastewater treatment. However, the wide application of AD has always been limited due to the low energy conversion efficiency of the sludge. To overcome this issue and enhance methane production, exogenous nano materials have been recommended after effective trials.

In relation to microorganisms, there are two aspects to the impact of iron: (i) it serves as an essential trace element for anaerobe microbes and improves competition with sulphate reducing bacteria (SRB) leading to the growth and reproduction of methane producing microbes; (ii) activities of the enzymes involved in methanogenesis and acidogenesis can be stimulated by iron due to its ability to improve basic elements in metallo-enzymes. The above-mentioned analysis strongly suggests that both metabolically and technically, enhancement of AD by nano-ions is feasible. Energy recovery through iron-based anaerobic digestion is a sustainable and promising strategy that covers many cross disciplinary fields. This technique can result in a novel industrial chain because it can interlink wastewater treatment, the steel industry and energy generation.

## 11. Future Prospects

For qualitative and quantitative production of biogas, nanoparticles can bring a revolution to meet the future needs of energy. The process of biofuel production can be altered by using various nanomaterials in various ways, such as by improving the stability of cellulose enzymes, enhancing the catalytic production of biohydrogen, and improving biological and chemical digestion. This influence of nanoparticles on the process is determined by their distinct catalytic activity based on structure, shape and size which is complementary to the relevant process. For understanding the basic mechanism of alteration in the production process and stability of the related protein carried out by nanomaterials, molecular investigations are helpful.

In order to achieve cost effective biofuels production, the cost involved in the synthesis of nanoparticles must also be taken into consideration. This cost can alter the overall process of biofuel production. Synthesis of cost effective nanomaterials can make the process of biofuel economically viable. On a brief note, we conclude that this area of research with a lot of potential still has some challenges that need to be overcome; (i) nanomaterials with controlled catalytic properties should be synthesized in order to improve the production process, (ii) economically viable nanoparticles, (iii) compatibility rate among microorganisms, involve enzymes and nanomaterials, and (iv) a molecular level understanding of the mechanism involving nanomaterials and proteins. The validation of optimal operational parameters for maximizing the energy output after the application of iron to the aforementioned organic wastes still needs further study. Furthermore, residual iron might be used to settle phosphate in the form of a high value potential phosphate compound called vivianite.

**Author Contributions:** X.L., Y.Z. and F.Y.H. reviewed the manuscript.; S.F. contributed in conception, designing and drafting of the review.; S.M. and X.L. contributed in preparing figures.; I.S. contributed in the interpretation of figures.; S.Z. and K.M. formulated tables.

**Funding:** This research received no external funding.

**Acknowledgments:** The current review article work was supported by Fundamental Research Funds for the Central Universities grant (No: lzujbky-2017-br01) and Gansu province People's Republic of China major science and technology projects (No: 17ZD2WA017). National Natural Science Foundation Grant (No: 31870082). The author also expresses colossal gratitude to COMSATS University Islamabad Pakistan and higher education commission Pakistan for awarding a scholarship under "international research support initiative program" grant number: IRSIP 39 BMS 35.

**Conflicts of Interest:** The authors declare no conflict of interest.

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
