# Peer review of "A Review on Nanoparticles as Boon for Biogas Producers—Nano Fuels and Biosensing Monitoring"

_applsci, doi:10.3390/app9010059_

Reviewer 1 Report

The manuscript entitled " A review on iron oxide nanoparticles as boom biogas producers- nano fuels and biosensing monitoring" is a well written comprehensive review on the field of nanotechnology applied for biofuel production and biosensing. Therefore, I would suggest to accept the aforementioned manuscript as it is for publication.

Author Response

Thanks for your time for reviewing our manuscript.

Reviewer 2 Report

The review article is timely and contains a good deal of information. However, I have concerns with regards to the quality of the presentation as highlighted below:

1- The title focuses on iron nanoparticles. But the article reviews the literature for a large number of nanoparticles. So, the authors should either consider changing the title to indicate that the review is not only for iron nanoparticles. OR  limit the scope of the review to iron nanoparticles in order to match the title.

2- The flow of the paper is confusing. The authors jump between different subject and there is no cohesion or a specific flow that make sense. I highly recommend reorganizing the article to ensure that there is a better flow of information.

3-  In line 55, Figure 3 is cited. But this figure does not belong to the text in lines 53-55. Please check that he figures cited align with the information in the text. 

4- Line 60: "Feeding iron nanoparticles to the bacteria which degrade organic matter stimulates its activity which reduces the emission of greenhouse gases, increase biogas production, renewable energy source  development." How is that? increase in biogas means increase in CH4 and CO2 production, both are greenhouse gases. Please try to re-write in order to eliminate contradictions. 

5- Line 68, "highest biggest production ever reported". Pleas reconsider re-writing this. Highest biggest does not sound correct.

6- In general, please consider proofreading the article and correct grammatical mistakes.

7- The resolution of many of the Figures presented in the article is low. Please find a way to increase the resolution of the figures.

8- Table 3 presentation is poor. Please consider changing format, merging cells, organizing the in a better way. This is an important table that could be presented in a much better way.

Author Response

Comments: Reviewer 2

1: The title focuses on iron nanoparticles. But the article reviews the literature for a large number of nanoparticles. So, the authors should either consider changing the title to indicate that the review is not only for iron nanoparticles. OR limit the scope of the review to iron nanoparticles in order to match the title. 

Response:Thank you, we have changed the title from specific to general yellow highlighted in the revised version. 

2:  The flow of the paper is confusing. The authors jump between different subject and there is no cohesion or a specific flow that make sense. I highly recommend reorganizing the article to ensure that there is a better flow of information. 

Respected sir/madam we have addressed this issue and we reorganized some parts of the manuscript according to yours instruction and are yellow highlighted in the revised manuscript.

3: In line 55, Figure 3 is cited. But this figure does not belong to the text in lines 53-55. Please check that he figures cited align with the information in the text. 

Thank you sir/ madam for yours comment we have removed the figure 3 from the text because no relation with text. 

4:  Line 60: "Feeding iron nanoparticles to the bacteria which degrade organic matter stimulates its activity which reduces the emission of greenhouse gases, increase biogas production, renewable energy source  development." How is that? increase in biogas means increase in CH4 and CO2 production, both are greenhouse gases. Please try to re-write in order to eliminate contradiction  

Thank you, the correction made and yellow highlighted (page 3, in Lines from 82 to 88) in the new draft of the manuscript.   

5:  Line 68, "highest biggest production ever reported". Pleas reconsider re-writing this. Highest biggest does not sound correct.

Respected sir/madam the correction made and is yellow highlighted (page 4, in Lines 113,114) In the revised draft. 

6: In general, please consider proofreading the article and correct grammatical mistakes.

Thank you, the proofreading and grammatical improvement were made. 

7:  The resolution of many of the Figures presented in the article is low. Please find a way to increase the resolution of the figures. 

Thank you, the resolution of the figures improved in the revised manuscript.

8: Table 3 presentation is poor. Please consider changing format, merging cells, organizing the in a better way. This is an important table that could be presented in a much better way. 

Respected sir/madam the presentation of the table 3 improved changed the format and organised according to the journal standard.

Reviewer 3 Report

The manuscript deals with an interesting subject, as it the application of nanoparticles and its effects on different processes. However, the concatenation of subjects is not well conducted seeming different sections with no reasonable relation exception of  that given in the title of the manuscript.) 

The introduction section (until L 108) should be completely rewritten because the English style is poor and makes difficult for the reader to follow the ideas expressed by the authors.

From section 1.1 to the end the manuscript is much clearly written and the quality is higher making the subject more interesting.

My recommendation is to completely rearrange the first part of the document. There are several figures associated with this first section which need deeper comments.

Please make sure Table 3 follows the journal standards.

Increase the quality of Scheme 1

Please check there is an inconsistency in L 143-145, and L 361-362

L 258, meaning not clear "Sulpher and Anaerobic digestion?" The whole phrase is lack of meaning

L 408, abbreviation does not keep relation with meaning "FOS/TAC" (volatile organic acids/total organic carbonate)

Section 1.8 is not well concatenated with the text. There is a need of giving a general outline in the introduction section to describe all subjects authors want to deal with, the same happens with the section "Application of Biosensors in Biogas Monitoring", it is completely out of place, and this is a problem related with the way the introduction was performed. There is also need of introducing a small comment in the previous section that leads the reader to the following one. This manuscript is written in a way that the different sections are not at all related unless the reader make an effort to find common aspects

I guess the text regarding section 4 is missing, because there is only two lines of text, and even a figure is presented, I guess regarding this section.

Author Response

Comments: Reviewer: 3

1: The introduction section (until L 108) should be completely rewritten because the English style is poor and makes difficult for the reader to follow the ideas expressed by the authors.

Thank you, we fully revised introduction portion and corrections/editing are yellow highlighted.

2: There are several figures associated with this first section which need deeper comments.

Thank you sir/madam we have revised the first section according to yours instruction and removed some figures and addressed the remaining in detail in the revised draft.

3: Please make sure Table 3 follows the journal standards.

Thank you, the table 3 is reformatted according to the journal standard.

4: Increase the quality of Scheme 1.

Respected sir/madam, the scheme 1 is removed due to copyright permission.

5: Please check there is an inconsistency in L 143-145, and L 361-362.

Thank you, the correction made, the phrase we rearranged according to yours instruction and yellow highlighted (page 8, 14 lines 205,206,421,422 and 423) in the revised manuscript. 

6: L 258, meaning not clear "Sulpher and Anaerobic digestion?" The whole phrase is lack of meaning.

Respected sir/madam, the phrase rearranged and yellow highlighted (page 11, lines 317,318) in the revised version. 

7: L 408, abbreviation does not keep relation with meaning "FOS/TAC" (volatile organic acids/total organic carbonate).

Thank you, the abbreviation corrected (page, line 62, 63) and yellow highlighted in the revised manuscript.

8: Section 1.8 is not well concatenated with the text.

Thank you, we have rearranged the portion.

9: Application of Biosensors in Biogas Monitoring", it is completely out of place, and this is a problem related with the way the introduction was performed.

Thank you, correction are made and the biosensors part is addressed in introduction and the introduction we rewrite yellow highlighted in the revised version(page 3, 53 to 81).

10: I guess the text regarding section 4 is missing, because there is only two lines of text, and even a figure is presented, I guess regarding this section.

Thank you, the section we removed. 

Round  2

Reviewer 2 Report

Thank you for the efforts to address my previous comments.

The only thing left is the language. I still find some grammatical errors in the manuscript and I recommend that the authors proofread the manuscript and correct the mistakes.

Author Response

Reviewer 2:We are thankful for the reviewer’s comments.

1: The only thing left is the language. I still find some grammatical errors in the manuscript and I recommend that the authors proofread the manuscript and correct the mistakes.

Response: as the reviewer suggested, we did proofreading and tried to minimize the grammatical mistakes and did corrections.

Reviewer 3 Report

L 52. The proper Word is not conclusion, changed to "the assessment of anaerobic digestion may be based on…." or any similar phrase

Introduce a paragraph explaining the aim of this review and the aspects that will be cover at the end of L 146 and previous to section 1.1, this way a concatenation of subjects is provided. 

Corrections performed by the authors were in accordance with suggestions preformed but a link between the different sections is still needed

Author Response

Reviewer 3:We are thankful for the reviewer’s valuable comments. The followings are our point-by-point responses:

1: L 52. The proper Word is not conclusion, changed to "the assessment of anaerobic digestion may be based on…." or any similar phrase

Response: as the reviewer suggested, we did the changes and changed the phrase according to reviewer instruction.

2: Introduce a paragraph explaining the aim of this review and the aspects that will be cover at the end of L 146 and previous to section 1.1, this way a concatenation of subjects is provided.

Response: thank you sir/madam for yours comments. We did the changes and reshuffled the paragraph of the aim of the review according to yours instruction and placed it before section 1.1.

3: Corrections performed by the authors were in accordance with suggestions preformed but a link between the different sections is still needed.

Response: We are thankful for the reviewer’s valuable comments. We did the changes to link different sections and reshuffled some sections in the revised version of the manuscript.
